# Rotaviruses and Noroviruses as Etiological Agents of Acute Intestinal Diseases of Ukrainian Children

**DOI:** 10.3390/ijerph19084660

**Published:** 2022-04-12

**Authors:** Serhii O. Soloviov, Tetiana S. Todosiichuk, Olena V. Kovaliuk, Gabriel M. Filippelli, Olena P. Trokhymenko, Iryna V. Dziublyk, Zachary A. Rodd

**Affiliations:** 1Department of Virology, Shupyk National Healthcare University of Ukraine, 04112 Kyiv, Ukraine; solovyov.nmape@gmail.com (S.O.S.); kovaliukolena@gmail.com (O.V.K.); trokhimenko@ukr.net (O.P.T.); idziublyk@ukr.net (I.V.D.); 2Department of Psychiatry, National Technical University of Ukraine “Igor Sikorsky Kyiv Polytechnic Institute”, 03056 Kyiv, Ukraine; todosiichuk.ts@gmail.com; 3School of Medicine, Indiana University, Indianapolis, IN 46240, USA; gfilippe@iupui.edu

**Keywords:** norovirus, rotavirus, infant, hospitalization, geographic distribution

## Abstract

(1) Background: Rotavirus and norovirus infections are the primary viral causes of childhood diarrhea. In Ukraine, the diarrhea-linked infant mortality rate is low, but the number of children infected is quite high. This study examined the rates of rotavirus and norovirus infections throughout Ukraine. (2) Methods: Fecal samples for children admitted to hospitals in six Ukrainian cities (Kyiv, Lviv, Sumy, Odesa, Kharkiv, and Uman) were tested for the presence of rotavirus and norovirus. (3) Results: The overall rate of hospitalized children suffering from diarrhea with confirmed presence of rotavirus or norovirus in fecal samples was significant (20.67% and 27.94%, respectively). Samples obtained from children from Lviv had significantly higher rates of the viruses, and Kyiv and Uman had significantly lower rotavirus or norovirus detection levels than expected. (4) Conclusion: Childhood diarrhea impacts Ukraine significantly. The economic and societal effects of the failure to address this public health issue are indicated by the hospitalization rate of children with preventable illnesses. The geographical disparities in Ukraine for child hospitalizations caused by rotavirus and norovirus infections could result from environmental (sanitary factors or water purity issues) or social factors. Further research is needed to completely characterize infant viral infections in Ukraine.

## 1. Introduction

There are 1.7 billion cases of ‘diarrhea’ in infants reported to the World Health Organization annually [1]. The two most common viral infections that produce diarrhea in infants are rotavirus and norovirus. Prior to the development of a rotavirus vaccine, there was an estimated 111 million yearly cases of rotavirus infection in infants (<5 years) that required home care (typically rehydration; [2]). There were also 25 million yearly clinical visits and over 2 million hospitalizations to treat rotavirus infections in infants [2]. Ultimately, rotavirus infections in infants resulted in a yearly mortality rate of between 352,000 and 592,000 deaths [3]. A consistent finding was that the infection rates for children in developed and developing nations were equivalent, but the burden of infant mortality was primarily in developing nations [4]. Rotavirus infections were so common prior to the development of a rotavirus vaccine that virologists concluded that all infants had been infected at least once by the rotavirus prior to the age of five [2,4].

The majority of diarrheal incidents in infants (49–67%) are caused by acute intestinal infections (AII). Norovirus infections are the second most common cause of AII and virus-induced childhood diarrhea [5]. However, at least eight different families of viruses result in the development of acute diarrhea in humans—*Reoviridae* (genus *Rotavirus*), *Caliciviridae* (genus *Norovirus*, *Sapovirus*), *Adenoviridae* (genus *Mastadenovirus*), *Astroviridae* (genus *Astrovirus*), *Picornaviridae* (genus *Enterovirus*), *Parechovirus*, *Kobuvirus* (agent of *Aichi*), *Coronaviridae* (genera *Coronavirus*, *Torovirus*), *Parvoviridae* (genus *Bocavirus*), and *Picobirnaviridae* (genus *Picobirnavirus*). Over the past decade, viral etiology’s nosological structure of acute diarrheal diseases has undergone significant changes [6,7]. During the existence of the Soviet Union, infant cases of diarrhea in Ukraine were not commonly diagnosed through laboratory testing, but infection rates were estimated by the presentation of symptoms of different diseases or were considered simply as being caused by a disease of unknown etiology [8,9]. Following Ukrainian independence (1991), more modern techniques have been employed to determine the biological basis of AII in Ukrainian children.

The first widely used rotavirus vaccine (Rotarix-RV1, GlaxoSmithKlein, Brentford, UK) was introduced in 2010. Countries bordering Ukraine (Moldova) or other former members of the Soviet Republic (Armenia) have embraced the use of RV1 to great benefit to their nations. During the first year of compliant RV1 vaccination in Moldova, the rate of infants hospitalized with diarrhea with a primary diagnosis of rotavirus gastroenteritis dropped from 45% to 25%, and there was a further reduction during the 2nd year of vaccination [10] (14%). Following five years of an active RV1 vaccination program, the detected rate of hospitalized children with diarrhea caused by rotavirus gastroenteritis was reduced from 38% to 20% in Armenia [11]. In Ukraine, there is resistance to both child and adult vaccination. The rate of effective poliovirus vaccination for Ukrainian children during 2015 was 15% [12,13]. Very few Ukrainian infants (>2 years) were vaccinated against diphtheria, pertussis, and tetanus in 2016 (UNICEF). In 2018–2019, there was a sustained measles outbreak in Ukraine (53,000 and 57,000 cases, respectively [14]). It is estimated that less than 1% of Ukrainian children have received RV1 vaccination (0.15–0.6% [15]). The low RV1 vaccination rate in Ukraine is influenced by the fact that individuals mostly pay for it through private clinics. The failure to vaccinate has resulted in the annual hospitalization rate of infants for AII remaining constant at roughly 50,000 for the last decade [16]. Extrapolating from previous data sets [2], it is likely that the majority of Ukrainian infants will be infected by the rotavirus before the age of 2 [15].

The consistency of viral etiology of AII in Ukraine indicates that between 2006 and 2015, the rate of rotavirus infection in infants hospitalized with AII in Kyiv was between 39% and 42% [15,17]. Despite the low vaccination rate against rotavirus, the mortality rate of infant diarrhea is low in Ukraine because of advanced medical treatment (generally no annual deaths [15]). Previous examinations of viral-induced AII in Ukrainian infants have sampled primarily from Kyiv hospitals (and sometimes Odesa [15,18]). Each region of Ukraine has distinct factors that could influence the rate of viral-induced AII in infants. The current study examined the presence of noroviruses and rotaviruses in Ukrainian children hospitalized with AII in six cities (Kyiv, Sumy, Kharkiv, Lviv, Odesa, and Uman). The overall hypotheses tested were that rotavirus and norovirus infections would be the primary viral source of children hospitalized with AII in Ukraine and that there would be minor variations in the rates of rotavirus and norovirus infections in the different Ukrainian cities.

## 2. Materials and Methods

### 2.1. Patients and Setting

Subjects were restricted to children under three years of age who were hospitalized for AII. The hospitals were located in 6 cities (Kyiv, Odesa, Kharkiv, Sumy, Uman, and Lviv). The hospitals in Kyiv and Odesa were not enrolled in the WHO surveillance program used in past research projects. A map of Ukraine has been modified to provide basic information of the cities selected for study (Figure 1).

### 2.2. Enrollment, Consent, and Diagnosis Inclusion

Children were recruited into the project if they met the WHO standard protocol for determining the biological cause for AII. To be classified as suffering from AII, children needed to have three or more episodes of non-bloody diarrhea within a 24 h period prior to hospitalization, but not for longer than seven days prior to hospitalization. Clinical symptoms, treatment, and demographic information about participants were obtained from medical records provided by the participating hospitals and not directly from adult guardians (although medical information was obtained directly from adults presenting children to the hospital). Parents or guardians of the children with AII were approached to obtain consent for the children to be included in the study. As is typical for Ukraine, the willingness of individuals to participate in scientific research (particularly research involving children–parental consent) was low. No monetary reward was offered for participating in the study. The ethical commissions of all hospitals approved the research protocols. Similarly, the research was approved by the Scientific Ethics Boards of the Shupyk National Healthcare University of Ukraine and National Technical University of Ukraine “Igor Sikorsky Kyiv Polytechnic Institute” (KPI–Code of Honor 77).

The research team was able to obtain consent from 569 parents/guardians. The samples were obtained from all selected regions (Kyiv, Kharkiv, Sumy, Uman–100; Kharkiv–99; Odesa–70). Samples were obtained over a four-year period (January 2016–January 2020). We attempted to sample equivalently throughout the year, but there is a general reluctance of Ukrainian parents to consent to experiments performed on their children. The majority of the samples were obtained from Nov to Feb of each year. These months correspond with the peak in infection rates of rotavirus and norovirus in Ukrainian children [15]. Participating hospitals did not provide the research team with the exact percentage of individuals enrolled in the experiment. Estimates provided by two hospitals for a single year indicated that between 4–8% of parents/guardians of Ukrainian children hospitalized with AII agreed to participate in the study.

On 7 February 2008, the Ministry of Health of Ukraine registered the ‘Organization of laboratories in the study of material containing biological pathogens of I–IV groups of pathogenicity by molecular genetic methods’ (Resolution 22.06.99 N 1109). This was an all-encompassing law enacted to protect the Ukrainian society from unregulated research that could produce major health crises for the country. Any research conducted in Ukraine that violates this resolution’s regulations can result in civil and criminal actions against the scientists who have violated the laws. In conjunction with this resolution, there are complex laws for conducting research in Ukraine. For example, no biological-active samples of pathogenic group 3 or lower can be shipped out of Ukraine. We conducted all research adhering to the resolutions, laws, and the testing standards of Ukraine.

The project was an observational assessment study (documenting the rate of virus infection in hospitalized children in cities of Ukraine). A between-subject factor in the study was the location of sample source (various Ukrainian cities).

### 2.3. Laboratory Testing

Stool samples were collected within 24 h of hospital admission. All virological testing occurred in laboratories at KPI and Shupyk National Healthcare University of Ukraine. Stool samples were suspended in a phosphate-salt buffer (50% suspension). Samples were divided into three sterile plastic test tubes (Eppendorf, Hamburg, Germany) to allow for individual ELISA testing for presence of viruses. Samples were then immediately placed in a −20 °C freezer. All samples were collected following the rules and regulations of Ukraine and researchers maintained a ‘chain-of-custody’ of the possibly virally infected samples.

Samples were only allowed to be thawed once prior to virus detection. The determination of the antigens of rotaviruses, adenoviruses (40/41), noroviruses, and astroviruses was carried out by the method of Enzyme-Linked ImmunoSorbent Assay (ELISA, Helsinki, Finland) in the laboratory of the department of virology Shupyk NHUU. To avoid contamination, the researchers employed “WELLWASH 4MK-2” (“THERMO ELECTRON”, Helsinki, Finland) to maintain the integrity of the samples. Diagnostic test systems from the firm R-BIOFARM AG (Darmstadt, Germany) (RIDASCREEN Rotavirus, RIDASCREEN Norovirus, RIDASCREEN Adenovirus, and RIDASCREEN Astrovirus), which passed registration in Ukraine, were used in the experiment. The methods used to perform the ELISA analyses were obtained directly from the manufacturers of the ELISA kits. Antigen reactions in the ELISA kits were determined by two-wave spectrophotometer “MULTISCAN ASCENT” 450/820 nm (“THERMO ELECTRON”, Helsinki, Finland).

Informativeness and the diagnostic value of the determination of virus antigens in the feces were analogous to the determination of an increase in the caption of the specific immunoglobulins IgA and IgG in the blood serum. Clinical, coprological and bacteriological studies of children were achieved at the same time. The complexity of the elapsing of acute gastroenteritis was evaluated according to 0–20 marks of the Vesikari scale. According to Russaka recommendations, the estimation of the scale “7 and less marks” characterized a low degree of acute gastroenteritis, “8–10 marks”—moderate to high degree, “11–13 marks”—high, “14 or more”—very high degree. Statistical data analysis was achieved with the use of the program Statistica 6.0 and/or SPSS version 8.0 (SPSS Inc., Chicago, IL, USA).

Statistical data analysis was achieved with the use of the program Statistica 6.0 and/or SPSS version 8.0.

## 3. Results

### 3.1. Viral Detection

Stool samples (collected within 24 h of hospitalization) obtained from 569 Ukrainian children with a primary diagnosis of AII indicated that the presence of rotavirus (20.7%) and norovirus (27.9%) was common (Figure 2). Detection of antigens for one of the five viruses that cause gastroenteritis (rotaviruses, adenoviruses, noroviruses, and astroviruses) occurred in 67.8% of stool samples tested. The detection of astroviruses (14.2%) was more common than the detection of adenoviruses (5%) in tested stool samples. There were correlations between the age of an infant and the detection of specific antigens of viruses that cause gastroenteritis (AII).

In young children (0–6 months of age), astroviruses were the primary agents of diarrhea of hospitalized Ukrainian children. Noroviruses (Figure 2) and adenoviruses were the most common viruses detected in Ukrainian children between the ages of 6 and 12 months. Rotaviruses emerged as the primary viral infection of AII in hospitalized Ukrainian children 12–36 months of age, but the presence of noroviruses was also common (Figure 2). Currently, we are unable to provide additional information about the data collected because of the Russian Invasion of Ukraine.

The average age of children surveyed in this study was 18.7 ± 1.35 months. Since the range of children included in the present study was 0–36 months, the data indicate that hospitalization for AII in Ukrainian children was normally distributed. Ukrainian children infected with rotaviruses were hospitalized sooner after the observation of any symptoms by parents/guardians than children with antigens for noroviruses detected in stool samples (Table 1). Confidence interval analyses (ratio level statistical analyses were not performed because of reliance on parent/guardian self-reports, multiple incidents of antigens of multiple viruses being detected in a single stool sample, and potential selection bias of hospitalization) revealed that Ukrainian children with detected antigens for rotaviruses were hospitalized sooner after the observation of symptoms than children with detected antigens for noroviruses (Wilk’s Likelihood Analysis L[θ] *p* = 0.002; Prθ,*φ* = 24.5; *p* < 0.001). Examining the percentage of viral antigen-positive stool samples as a function of the source (Ukrainian city) revealed distinct patterns of infection rates for children hospitalized with AII. Non-parametric statistical analyses were performed (χ^2^). For all stool samples collected (collapsed across age of child), the percentages of rotavirus antigen-positive samples (Figure 3) from Lviv (44.3%) and Odesa (45.4%) were statistically greater than that observed for the whole nation of Ukraine (20.67%; Lyiv *p* < 0.001; Odesa *p* = 0.005). In contrast, the detecting presence of rotavirus in children from Uman was significantly lower than the overall national mean (χ^2^ = 15.13; *p* = 0.01).

Positive antigen detection for noroviruses was also differentially expressed between communities in Ukraine (Figure 4). Again, the percentage of antigen-positive samples in Lviv was elevated compared to the other tested Ukrainian communities (χ^2^ = 16.3; *p* = 0.003). The number of positive norovirus antigen stool samples detected in Kyiv was significantly lower than the Ukrainian national average (χ^2^ = 10.7; *p* = 0.04). Examining the total number of rotavirus and norovirus antigen-positive stool samples for the six Ukrainian communities indicates that Lviv accounted for a significantly higher number of rotavirus- or norovirus-associated AII (37% of all cases). In fact, Lviv was the only community where infants hospitalized with AII had positive antigen detection for both noroviruses and rotaviruses. In contrast, Uman (8%) and Sumy (9%) had a significantly lower number of rotavirus- or norovirus-associated AII.

### 3.2. Characterization of AII

The frequency of the development of basic symptoms (hyperthermia, vomiting, diarrhea) in patients with rotavirus and norovirus infections in accordance with the Vesikari scale is illustrated in Table 1. Severe disease in children with rotavirus infection (according to the estimation of the general state of the child by a doctor) was, on average, for a period of 3.5 ± 0.34 days, which was statistically lower (*p* < 0.05) than norovirus infection (4.6 ± 0.4 days).

In the majority of patients (68.2%) with rotavirus and norovirus infections at the beginning of the hospitalization, the presence of three basic symptoms was described: vomiting, diarrhea, and hyperthermia. However, there were significant differences in the expression of symptoms between rotavirus- and norovirus-infected children (Table 1). Besides the symptoms of gastroenteritis, the clinical manifestations of upper respiratory tract symptoms were described: acute rhinitis, pharyngitis, tracheitis, and tonsillitis (Table 1). The symptoms of respiratory distress in hospitalized children differed between rotavirus-and norovirus-infected children (Table 1).

### 3.3. Detection of Bacterial Infection in Conjunction with Viral Infection

Each hospital tested for the presence of multiple bacterial infections during hospital admittance. All hospitals performed these tests, and the data were shared with the researchers. The researchers combined our viral testing with the data generated by the hospitals to determine if co-morbid bacterial infections were common in Ukrainian children hospitalized with AII. The Ukrainian hospitals tested for infection with the Salmonella, Klebsiella, Proteus, Staphylococci, and Enterobacter bacteria (Table 2). Assessment of co-morbid bacterial infection was also determined to eliminate a possible confounder in the study of AII severity (for example, children with antigen-positive rotavirus and positive Salmonella could have more serve symptoms than children with only antigen-positive rotavirus AII). Pathogenic flora was isolated from the defecations of 70.1% of children with rotavirus infection and from 65.2% of children with norovirus infection. In general, there was frequent co-morbid viral and bacterial infection in hospitalized children with AII (Table 2). We wanted to provide more information about bacterial infection for this publication, but the current war has prevented us current access to the data (in addition, there has been significant damage to some of the participating hospitals—the location of data storage).

As required by hospital protocol in Ukraine, all children hospitalized with AII were tested for the presence of the fungi Candida. The research team did not perform these tests, but the data were freely shared by the hospitals. There was a considerable co-morbidity rate between viral and Candida infection. Specifically, coprological testing revealed that in 36.4% of children with norovirus infection and in 19.2% of children with rotavirus, Candida infection was also present.

## 4. Discussion

The data indicate that viral-induced AII is common in hospitalized Ukrainian children (Figure 2, Figure 3 and Figure 4). Overall, the infection rate of young children (ages 0–36 months) in six Ukrainian communities indicated that there were age-specific (Figure 2) and geographical factors (Figure 3 and Figure 4) influencing the viral causes of AII. Stool samples obtained from very young children (0–6 months) hospitalized with AII indicated the presence of astroviruses and adenoviruses. In slightly older children (6–12 months), noroviruses were the predominant virus antigen detected. In children older than one year, rotaviruses were the most commonly detected viral antigen.

Breastfeeding (exclusively or majority) has been linked with a reduction in the rate of rotavirus infections in multiple studies [19,20]. In Ukraine, the rate of exclusively breastfeeding neonates (0–6 months) is only 19.6%, but over 70% of infants receive the majority of their nutrition from breastmilk for the first year of life. In a recent survey of breastfeeding activity in European countries, Ukraine was ranked the third highest for promoting, monitoring, and the rate of breastfeeding [21]. The emergence of rotavirus-infection-induced AII in Ukrainian children at one year of age (Figure 2) corresponds to a dramatic breastfeeding reduction in Ukrainian females [21].

The current reported rate of rotavirus infection (20.7%) is slightly lower than previous reports examining the rate of rotavirus infection in hospitalized AII children [15,18]. There are two factors that could explain the discrepancy: the age cohort of the sample and the time of sampling after hospitalization. Previous reports have examined the rate of rotavirus infection in Ukrainian children aged 0–5 years. The current study age cohort was from 0 to 3 years of age, and breastfeeding would have lowered the rate of rotavirus infection compared to studies examining children removed from the parent–child activity. Second, the current research collected stool samples within 24 h of hospitalization, while the other studies collected samples 48 h after hospitalization. Rotaviruses (and other viruses) are endemic in Ukrainian hospitals [22]. Therefore, children hospitalized in Ukraine may have been exposed to viruses targeted for research while being hospitalized.

Past studies examining rotavirus infections in Ukrainian children hospitalized with AII have been limited to the Kyiv and/or Odesa communities [15,18]. In general, the rate of viral infections for Kyiv and Odesa in these studies has been overlapping. The current data indicate that Lviv has a significantly higher level of antigen-positive stool samples in children hospitalized for AII for rotaviruses and noroviruses than the national average. Lviv is the largest western Ukrainian city and is along the borders of Poland, Slovakia, Romania, and Hungary. Lviv is on a critical trade route between Moscow/Kyiv and major European cities. It is possible that the role of Lviv in trade enhances the rate of rotavirus and norovirus infections in AII-hospitalized children. However, another possible factor enhancing the rate of viral infections in Lviv children hospitalized with AII is the water supply. Unlike all other Ukrainian communities included in the research project, the water supply of Lviv is mostly derived from untreated well water (85–92% of residents; [23]). The water quality of these wells is questionable, since the Western Bug River is one of the five most polluted rivers in Europe [24]. Recent studies have indicated that anthropogenic pollutants are well above EU standards, and the contaminated Western Bug River water inundates the wells of Lviv [25]. The Western Bug River is polluted by industrial, agricultural, and community elements, but the largest polluter is the outdated Lvivvodokanal wastewater treatment plants and the sludge processing sites of the facility [23]. Therefore, it is possible that the increased rate of rotavirus and norovirus antigen-positive stool samples in young children hospitalized with AII is linked to poor water quality.

Hospitalized children from Uman and Sumy were significantly less likely to have positive antigen rotavirus and/or norovirus fecal specimens compared to the other Ukrainian communities studied. Three factors could impact the rate of rotavirus and norovirus antigen-positive stool samples in children hospitalized with AII in these communities: environmental pollution, population density, and maternal behavior. Uman is located in central Ukraine. Uman was the smallest community selected for this study, and has low population density and a gardening research center (University of Gardening) that benefits from the low level of pollution and high-quality soil. Sumy is a medium-sized city, but has a low population density. Sumy is in the northeast portion of Ukraine and is along the border of Russia. Sumy is located away from other major cities (Kharkiv, Ukraine, and Kursk, Russia). Sumy, similarly to Uman, was regarded as having low environmental pollution (until recent industrial development in Sumy [26]).

Exclusive breastfeeding by Ukrainian mothers is higher in small/medium-sized cities like Uman and Sumy [27]. Low population density is associated with a lower rate of rotavirus and norovirus infection [28]. Although not testable in the context of the current study, regional differences in rotavirus and norovirus positive fecal specimens of hospitalized children with AII, which were suggested by the data, could be influenced by environmental, behavioral, and residential factors.

Viral gastroenteritis in young children is often associated with norovirus infection. An epidemiological survey of six independent states of the former Soviet Union (Ukraine, Republic of Moldova, Belarus, Georgia, Armenia, and Azerbaijan) revealed that in rotavirus antigen-negative fecal samples, noroviruses were detected in 21.8% of hospitalized children [28]. The results obtained in the current study are within the range of norovirus-induced childhood AII (27.92%; Figure 4).

There are some limitations in the data collected. First, the low participation rate could have produced a bias in the data set. Although the data obtained were from Ukrainian children hospitalized with AII, it is possible that there was a positive correlation between the severity of the child’s illness and willingness to participate in our research project. Second, we did not test for all the viruses known to cause childhood AII (saprovirus). Third, we had no way to directly assess the two primary factors that we linked to rotavirus and norovirus infection rates in our data set: breastfeeding and drinking water purity. Fourth, samples were not collected evenly throughout the year because of a lack of participation during certain months. This limitation reduced our ability to examine seasonal effects in our data. An additional limitation for this publication is that for safety reasons most of the Ukrainian collaborators had to flee their homes (except S.O.S) and were not allowed to physically be at their universities. Furthermore, we were unable to address some valuable comments from the reviewers because participating hospitals have received damage from the war (data storage) and some data may be permanently destroyed as the result of the war.

In conclusion, rotavirus and norovirus infection pose a significant health risk in young children in Ukraine. The failure to successfully implement rotavirus vaccination programs in Ukraine results in a significant health burden to Ukrainian children. Regional differences in the detection rate of rotavirus and norovirus antigen-positive stool samples suggest that multiple factors can influence the presence and severity (hospitalization inclusion of the current study) of viral infections in a developing nation. Active monitoring of public health factors needs to be supported in Ukraine, and an increase in collaborative projects between Ukraine and foreign researchers will benefit Ukraine and the world.

## Figures and Tables

**Figure 1 ijerph-19-04660-f001:**
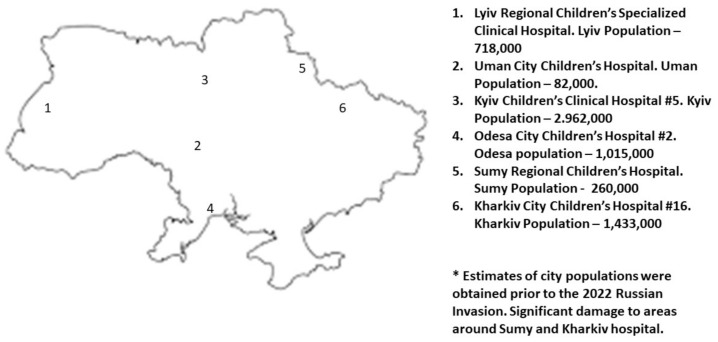
A map of Ukraine with the locations of the cities included in the study indicated.

**Figure 2 ijerph-19-04660-f002:**
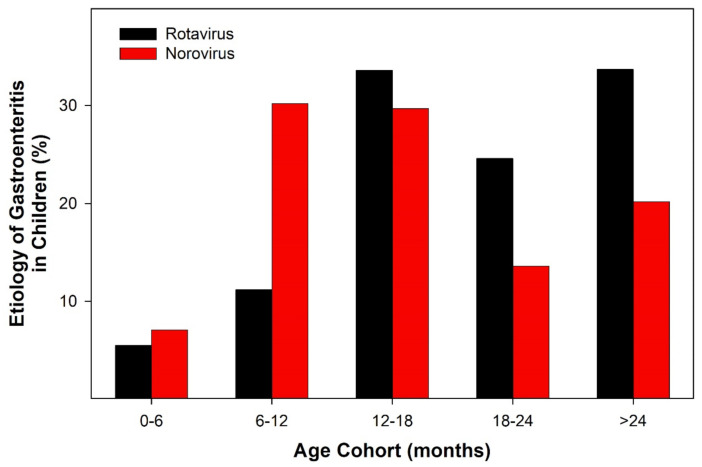
Depicts the percentage of antigen positive stool samples for rotavirus or norovirus obtained from young Ukrainian children (ages 0–36 months) hospitalized for AII as a function of age of the child.

**Figure 3 ijerph-19-04660-f003:**
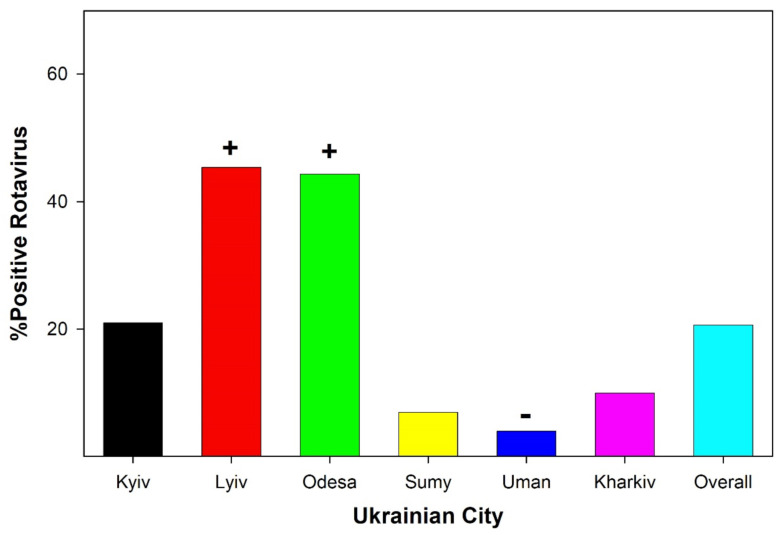
Depicts the percentage of antigen-positive stool samples for rotavirus obtained from young Ukrainian children (ages 0–36 months) hospitalized for AII as a function of the community that the fecal specimen was collected from. + indicates significantly higher rotavirus antigen samples compared to national average. − indicates significantly lower rotavirus antigen samples compared to national average.

**Figure 4 ijerph-19-04660-f004:**
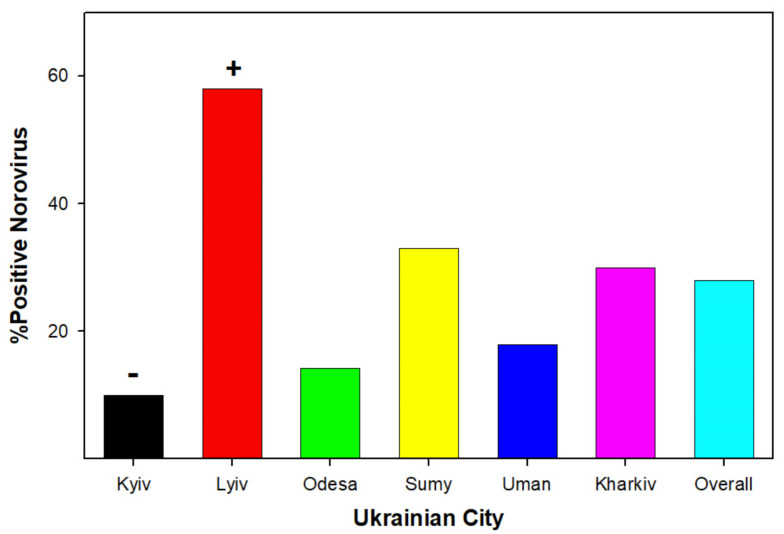
Depicts the percentage of antigen-positive stool samples for norovirus obtained from young Ukrainian children (ages 0-36 months) hospitalized for AII as a function of the community that the fecal specimen was collected from. + indicates significantly higher norovirus antigen samples compared to national average. − indicates significantly lower norovirus antigen samples compared to national average.

**Table 1 ijerph-19-04660-t001:** Basic symptoms of virus gastroenteritis in accordance with the Vesikari scale.

Clinical Manifestations	Rotavirus Infection	Norovirus Infection
The Duration of Diarrhea (Days)		
<1 day (0 marks)	0	8.7%
1–4 (1 mark)	50.0%	43.5%
5 (2 marks)	19.2%	21.7%
>5 (3 marks)	30.8%	26.1%
The greatest number of defecations during 24 h		
1–3 (1 mark)	26.9%	30.4%
4–5 (2 marks)	30.8%	34.8%
>5 (3 marks)	42.3%	34.8%
The duration of vomiting (days)		
None (0 marks)	11.5%	8.7%
1 (1 mark)	50.0%	34.8%
2 (2 marks)	23.1%	21.7%
>2 (3 marks)	15.4%	39.1%
The greatest number of cases of vomiting during 24 h		
None (0 marks)	11.5%	8.7%
1 (1 mark)	3.8%	4.3%
2–4 (2 marks)	57.7%	43.5%
>4 (3 marks)	26.9%	43.5%
Hyperthermia		
<37.1 (0 marks)	26.9%	30.4%
37.1–38.4 (1 mark)	34.6%	34.8%
38.5–38.9 (2 mark)	23.1%	26.1%
>38.9 (3 marks)	15.4%	8.7%
Dehydration		
None (0 marks)	42.3%	43.5%
1–5% (2 marks)	26.9%	17.4%
>5% (3 marks)	30.8%	39.1%
The treatment		
Not carried out (0 marks)	0	0
The rehydration (1 mark)	56.5%	39.1%
Hospitalization (3 marks)	53.5%	60.9%

**Table 2 ijerph-19-04660-t002:** Frequency of the isolation of bacteria from the defecations of children with rotavirus and norovirus gastroenteritis.

Agent	Rotavirus	Norovirus
*Salmonella*	0.5	0.14
*Klebsiella*	0.23	0.27
Proteus	0.08	0.09
*E. aeruginosa*	0.12	0.32
*Staphylococcus aureus*	0.15	0.05
Not isolated	0.27	0.36

## Data Availability

Data is available upon request.

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
