# Peer review of "Rotaviruses and Noroviruses as Etiological Agents of Acute Intestinal Diseases of Ukrainian Children"

_ijerph, 2022, doi:10.3390/ijerph19084660_

Round 1

Author Response

Response to Reviewer 1

We thank the reviewer for their helpful comments.  We are sorry that given the reality of the situation in Ukraine right now, we can not probably address some of the reviewer’s comments.  We hope that the reviewer understands this unique limitation.

General Comments

  1. We agree with the reviewer and have made sure that there is consistency in the manuscript to discuss rotavirus, norovirus, and then other diseases. We have changed the title also to reflect the correct order.

Introduction

  1. Duplicated virus – thank you for the catch. It has been fixed.
  2. Correct about the statement of the USSR period. We have added your suggestion.
  3. Thank you for the reminder that Ukraine does not support the Rotarix vaccine. This has been added to the manuscript

Methods

  1. We have added a map of Ukraine with the details of the cities and hospitals used in the current study.
  2. The data was collected from Jan 2016 – Jan 2020. It is difficult to obtained informed consent from parents to test their children for the presence of virus infection.  We agree that rotavirus and norovirus infections are seasonal.  We did not have an even distribution of samples throughout the year.  Despite multiple efforts to obtain samples throughout the year, we failed to obtain this goal.  However, all of the cities tested has a similar bias in the obtainment of samples and informed consent (Nov – Feb).  Thus, samples were obtained at the same time in all regions.
  3. The hospitals did not keep a recording of potential recruits to the experiment.
  4. On February 7th, 2008, the Ministry of Health of Ukraine registered the ‘Organization of laboratories in the study of material containing biological pathogens of I-IV groups of pathogenicity by molecular genetic methods’ (Resolution 22.06.99 N 1109). This was an all-encompassing law enacted to protect the Ukrainian society from unregulated research that could produce major health crises for the country. Any research conducted in Ukraine that violates this resolution's regulations can result in civil and criminal actions against the scientists who have violated the laws. In conjunction with this resolution, there are simple laws for conducting research in Ukraine.  For example, no biological-active samples of pathogenic group 3 or lower can be shipped out of Ukraine. The adenovirus detection we used in the current experiment adhered to this law (Adenovirus 40/41 was detected).  There was uncertainty about testing other types of Adenoviruses, so we only tested for the approved 40/41 strain.
  5. In Ukraine, participation in research studies is greatly reduced if individuals must record personal information. All symptoms and demographics were provided by the attending physicians.
  6. The bacterial testing was performed by the participating hospitals. We were informed that all Ukrainian hospitals received standardized ELISA kits to test for bacterial antigens.  These ELISA test kits were usually manufactured within the Ukrainian Ministry of Health or related companies.  We were instructed that the tests were reliable and accurate.
  7. Removed

Results

  1. The senior author misread the requirements of the journal. Results are now longer in bullet presentation.
  2. The sad reality is that we are unable to retrieve the additional data from KPI and Shupyk. KPI houses the Department of Military Training (KVP).  Most Ukrainian military leaders were trained at KVP and there are housing/dormitory units for military trainees.  KPI and Shupyk are main targets for shelling and bombings.  KPI has been affected by Russian attacks.  Academic professors are not allowed to venture into KPI.  Furthermore, Drs.  Todosiichuk, Kovaliuk, and Dziublyk have departed Kyiv and were not allowed to take items belonging to KPI (computers) with them.  Trokhymenko’s location is unknown currently.  Dr. Soloviov has stayed in Kyiv, but he is unable to go to Shupyk.  There is a real possibility that all of the hard data for this project will be lost.  If there is an occupation of Ukraine, the data will be lost.  Two of the hospitals involved in the study have experienced shelling (Sumy and Kharkiv).
  3. We cannot address the seasonality. This is a limitation of the study (which is no clearly indicated in the manuscript).
  4. We believe the Chi Square analyses are necessary.
  5. Viral coinfections were not the focus of the manuscript. We do recognize the value of this research.  Again, we are not able to get the data.  The American collaborators assisted in performing the ELISA viral analysis performed at KPI.  Because of regulations associated with Ukrainian laws and resolutions, the American collaborators are not allowed to maintain copies of certain data in America.
  6. Thank you for understanding this ‘lost in translation’ issue. We have edited to your suggestion.
  7. We cannot access the data. We are uncertain that we will ever have access to the data again.
  8. Thank you for noticing this error. The ‘little’ has been changed to ‘frequent’.
  9. We can not get the candida data in time for resubmission. We are sorry about this inability.

Discussion

  1. Actually, we disagree with the reviewer on their ‘age cohort effect’ on the lower rotavirus rate observed in our study. In our study 28% of our subjects were <18 months old.  In other studies using the age parameter of 0-5 years, most report less than 15% of subjects < 18 months old.  Therefore, our population included a significantly higher portion of children (by age) that expressed lower levels of rotavirus infections.  We have addressed the seasonality issue, and we are unable to statistically analyze this factor in our current data.
  2. We agree that the sapovirus is present in 5-6% of AII children. We don’t believe that excluding sapovirus analysis greatly impacted our study. Pitkänen O, Markkula J, Hemming-Harlo M. Sapovirus, Norovirus and Rotavirus Detections in Stool Samples of Hospitalized Finnish Children With and Without Acute Gastroenteritis. Pediatr Infect Dis J. 2022 Feb 18. doi: 10.1097/INF.0000000000003493. Epub ahead of print. PMID: 35185141.
  3. There is migration of Ukrainians into Poland through Lyiv. This also means more cross traveling between Poland and Ukraine at the transit point of Lyiv.  It is hard to link migration with the current study with a degree of certainty.
  4. The ELISA tests performed are well established. The RT PCR tests are an advancement in detecting viral presence in samples.  We worked within the limitations of conducting research in Ukraine.

Reviewer 2 Report

The study: Noroviruses and Rotaviruses as Etiological Agents of Acute Intestinal Diseases of Ukrainian Children by Soloviov and colleagues performed a survey in 569 children to determine the presence of viral antigens (norovirus, rotavirus, adenovirus, and astrovirus) by commercial assays.

The aim of the study was to determine the frequency of these viruses in children with acute gastroenteritis (GI) from fecal samples collected in distinct regions (Kyiv, Kharkiv, Sumy, Uman, Kharkiv, and Odesa) of Ukraine.

The manuscript was poorly organized, the abstract lacks minimal information about what was done. The introduction has many obscure and flawed sentences and is not informative regarding the objective of the study. The methodology has paragraphs and sentences unreadable. Results were presented in a way that it is hard to understand the experimental approach. The discussion section is too long and fails to pinpoint the main features regarding the problem of GI in children.

In addition, the manuscript has grammatically wrong sentences, poorly structured sentences, besides the references need to follow the format of this journal.

Minor concerns:

Page 1 lines 1-24: Provide more accurate information on the total number of individuals included in this study. Also describe the methodology used to detect norovirus, adenovirus, astrovirus, and rotaviruses. Is there any difference regarding norovirus and rotavirus frequencies among the cities?

Page 1 line 23: ...environmental (proximity to national borders) or other factors… How the proximity to national borders can affect the raw number of children affected by gastroenteritis?

Page 1 line 12: ...Norovirus infections are the second most common cause of AII and virus-43 induced childhood diarrhea… Then what is the main cause of diarrhea in children?

Page 1 lines 45-50: viral genera and families must be in italics

Page 2 lines 50-64: These sentences should be in the same paragraph. These sentences are mostly useless to the subject of this study. Please provide information about the size of cities included in the study also provide information with relevant features such as economic index, health, and sanitary indexes, among others

Page 2 lines 73-74: I guess you what to say that parents are reluctant to vaccinate their children in Ukraine

Page 2 line 94: Please give more details about these characteristics between these cities that may have some impact on the frequency and status of AII among children.

Page 2 lines 97-98: This sentence is bizarre, perhaps you want to say that more studies are needed in order to explore in more detail or use another approach to determine the prevalence of norovirus and rotavirus in Ukraine

Page 3 lines 111-112: Please, exclude this sentence. This is another inappropriate sentence in the manuscript. Children are naturally curious about science. Besides, the typification of Ukrainian or any other ethical group as being less prone to cooperate with science must be based on facts and not in preconception. This sentence adds nothing to the study and must be removed.

Page 3 lines 123-126 and lines 130-133: These sentences describing the methodology are unreadable.

Page 4 line 151: Perhaps you want to say clinical diagnoses

Page 4 line 159: Although mentioned in the text adenoviruses were not shown in figure1

Page 6 lines 215-219: This sentence is confusing, rephrase it to make the message clear.

Page 8 line 266: (Table 2) Define NVI and RVI

Page 8 lines 284-285: Discuss and describe what are the main factors and how they may affect the frequency of noroviruses and rotaviruses.

Page 9 lines 309-310: Not sure if you can detect viral antigens within 24hs after the hospital admission and potential exposure to norovirus and rotaviruses.

Page 9 lines 331-332: I would say it is highly likely that poor quality of drinking water and inappropriate treatment of sewage have a high impact on the frequency of AII cases associated with norovirus and rotavirus.

Author Response

Response to Reviewer 2

We appreciate the comments from the reviewer.

General Comments

The manuscript was processed with the leading grammar software (Grammarly).  The manuscript was also assessed with a business grammar software program (Rodd is the founder of RRRauser Analytics and has access to Vulcan program assistance).  No other reviewer expresses the concerns of this reviewer.  There may have been wording compromises between the Ukrainian and American collaborators, but that is it.

The results section should not have been bulleted.  This was a misreading of the journal’s requirement.  References are corrected.

Abstract Suggestions –

Response – IJERPH only allows 200 words in the abstract.  The journal actively discourages including unnecessary method details in the abstract. 

Proximity to national borders – All of the other reviewers understand that this refers to migration or travel between countries.

Introduction

Reviewer - Norovirus 2nd most common cause of AII. 

Response - The statement is not inaccurate.  Rotavirus is the most common viral cause of AII in children.  This is indicated in the paragraph above this statement; ‘Rotavirus infections were so common prior to the development of a rotavirus vaccine that virologists concluded that all infants had been infected at least once by the rotavirus prior to the age of five2,4’ We believe that this is a clear indication what is the most common viral cause of diarrhea in children.

Reviewer - Page 1 lines 45-50: viral genera and families must be in italics

Response – Corrected.

Reviewer Page 2 lines 50-64: These sentences should be in the same paragraph. These sentences are mostly useless to the subject of this study. Please provide information about the size of cities included in the study also provide information with relevant features such as economic index, health, and sanitary indexes, among others

Response – We disagree with the reviewer.  I am not sure why the reviewer would not want us to emphasize the importance of studying Ukraine.  We have added a map of Ukraine with the cities and hospital information.

Reviewer - Page 2 lines 73-74: I guess you what to say that parents are reluctant to vaccinate their children in Ukraine

Response – We are not sure how to respond to this critique.  If desired, we could provide the readers with five pages of the history of vaccination reluctancy in Ukraine.  We can discuss the Russian-funded disinformation campaign about vaccination in Ukraine.  We will be willing to discuss the Russian-funded ‘grandmother/healer’ television program that weekly discussed infant deaths caused by the ‘poisons in their vaccines’.  However, YES, Ukrainians are more likely to have fake vaccination records than they are to be vaccinated.

Reviewer- Page 2 line 94: Please give more details about these characteristics between these cities that may have some impact on the frequency and status of AII among children.

Response – This is clearly mentioned in the discussion.  This is the proper way to organize a manuscript.

Reviewer - Page 2 lines 97-98: This sentence is bizarre, perhaps you want to say that more studies are needed in order to explore in more detail or use another approach to determine the prevalence of norovirus and rotavirus in Ukraine

Response –The sentence is not ‘bizarre’.  The sentence states that almost no Ukrainian children receive the RV1 vaccine (between 0.15-0.6%).  It is a grammatically correct sentence.  However, to appease the reviewer we have simplified the wording of the sentence.

Reviewer - Page 3 lines 111-112: Please, exclude this sentence. This is another inappropriate sentence in the manuscript. Children are naturally curious about science. Besides, the typification of Ukrainian or any other ethical group as being less prone to cooperate with science must be based on facts and not in preconception. This sentence adds nothing to the study and must be removed.

Response – We have no idea what the reviewer is referring to.  This is a willful misreading of the sentence.  Children under 3 cannot volunteer for the study.  Informed consent came from the parents.  These sentences were written by the Ukrainian researchers.  We are not sure how the reviewer can speak against Ukrainians.  This is a grossly inappropriate comment by the reviewer.  We have edited the single possible confusion that can be willfully misinterpreted.

Reviewer - Page 3 lines 123-126 and lines 130-133: These sentences describing the methodology are unreadable.

Response – The sentences are readable and accurate.

Page 4 line 151: Perhaps you want to say clinical diagnoses

Response - fixed

Page 4 line 159: Although mentioned in the text adenoviruses were not shown in figure1

Response – There was no mention that the adenovirus infection rate was depicted in Figure 1.  The manuscript clearly states - Noroviruses (Fig. 1) and adenoviruses were the most common viruses detected in Ukrainian children between the ages of 6-12 months.  This means that Noroviruses were depicted in Figure 1, but there is no indication that adenoviruses were depicted.

Reviewr - Page 6 lines 215-219: This sentence is confusing, rephrase it to make the message clear.

Response – We disagree with the reviewer’s opinion.

Reviewer - Page 8 line 266: (Table 2) Define NVI and RVI

Response - Corrected

Reviewer - Page 8 lines 284-285: Discuss and describe what are the main factors and how they may affect the frequency of noroviruses and rotaviruses.

Response – All other reviewers believe we have performed this task.

Reviewer - Page 9 lines 309-310: Not sure if you can detect viral antigens within 24hs after the hospital admission and potential exposure to norovirus and rotaviruses.

Response – Our samples were collected within 24 hrs of admission.  The time of performing the tests is irrelevant.  As clearly indicated in the manuscript, rotavirus and norovirus are endemic in Ukraine.  It is possible that children can become infected in the hospital.  Earlier sampling results in a more accurate assessment of the rate infection prior to hospitalization.

Reviewer - Page 9 lines 331-332: I would say it is highly likely that poor quality of drinking water and inappropriate treatment of sewage have a high impact on the frequency of AII cases associated with norovirus and rotavirus.

Response – This statement can be interpreted as having negative connotations against Ukraine.  We agree that the high rate of infection in Lyiv may be based upon poor drinking water.

Reviewer 3 Report

Firstly, I would like to congratulate the authors for submitting their work. This study examined the rates of viral infections in fecal samples of children from six cities of Ukraine. The overall rate of hospitalized children suffering from diarrhea with the presence of rotavirus or norovirus in fecal samples was significant (20.67% and 27.94%, respectively). Also, the samples from children from Lviv had significantly higher rates of the presence of the viruses.

The study has merit and will be of interest to our readers. However, my comments are appended below:

Introduction: This section is very lengthy. Please trim this section and remove unnecessary information.

-Please also provide a hypothesis in 2-3 lines at the end of this section.

Materials and methods: Please mention the study design in this section.

Results: well-written. The characterization of AII is very lengthy. Please reduce this section. Also, please avoid any duplication of findings in Table 1 and in the text.

Discussion: Please mention the limitations of this study in the discussion section.

Author Response

Response to Reviewer 3

General Response – We thank the reviewer for their support and helpful comments.  We have addressed your concerns and agree with their merit.

Reviewer’s Comment  - Introduction: This section is very lengthy. Please trim this section and remove unnecessary information.

Response: We agree with the reviewer.  We were concerned that many individuals would not understand the importance of Ukraine to Europe and the world.  Ukraine is an amazing country that has been ignored by the West (America).  Prior to the war, few Americans and other nationalities would be able to locate Ukraine on a map or understand the importance of this nation.  We have reduced some general statements in the introduction.

Reviewer’s Comment -Please also provide a hypothesis in 2-3 lines at the end of this section.

Response – Agree and added.

Reviewer’s Comment - Materials and methods: Please mention the study design in this section.

Response - Added

Reviewer’s Comment - Results: well-written. The characterization of AII is very lengthy. Please reduce this section. Also, please avoid any duplication of findings in Table 1 and in the text.

Response – We agree.  We have reduced the characterization of AII and removed redundant information in text.

Reviewer’s Comment - Discussion: Please mention the limitations of this study in the discussion section.

Response - Added

Reviewer 4 Report

In the current manuscript “Noroviruses and Rotaviruses as Etiological Agents of Acute Intestinal Diseases of Ukrainian Children” authors Soloviov et al. analyzed the norovirus and rotavirus from the fecal samples collected from across different cities from Ukraine. They found out 20.67% and 27.94% fecal samples were positive for rotavirus and norovirus, receptively. This study indicates the importance of implementing vaccination against these viruses may overcome the illnesses caused in young Ukrainian children’s.

This manuscripts is well written and presented.

Author Response

Response to Reviewer 4

We thank you for your comments.

To improve the Introduction we have reduced some of the tangential information and have corrected the error on the references presentation.

Reviewer 5 Report

Dear authors,
Good initiative. Please find below my comments.

Title: Start with rotaviruses for the sake of consistency. Throughout the text, you always mention rotaviruses first.

Lines 12-13: "This study examined the rates of viral infections throughout Ukraine" - you should mention which viral infections you intend to study

Line 23: you do not need to mention the national borders because it is confusing. For a non-Ukrainian, the relationship between the proximity to the borders and environmental issues is not apparent.

INTRODUCTION
Please review the referencing style. MDPI's is specific, and you can find it in the author's guidelines.

I will not go into detail in the introduction, but you must consider the following points:
1. Be brief, clear, and direct. You do not have to describe in detail vaccination if your target population is not immunized

2. Lines 44-50: not necessary. Since you have already started talking about rotavirus and norovirus, you do not need to mention other viruses. It would help if you started from the general information to the particular.

3. Focus on Ukrania. You can mention other areas, but the sooner you start talking about your study area, the better.

4. Separate the paragraph where you state your objectives.

MATERIAL AND METHODS
Lines 113-116: mention the number of the ethical approval letter for tracking purposes.

RESULTS
Do not present the results in bulleted paragraphs. Use normal text.

Figure 1's content will be better presented as two circular graphs with percentages. You can put them side by side as a figure with A and B.

I propose you present some graphs of co-infection (rotavirus-norovirus) if possible. That would be a piece of valuable information.

CONCLUSION
Line 358: instead of "poses" write "pose"

REFERENCES
There are some inconsistencies:
1. Use MDPI style
2. Why are some titles in capital letters?

Author Response

Response to Reviewer 5

We thank the reviewer for their helpful comments.  We appreciate the suggestions and have incorporated them in the manuscript.

Reviewer’s Comment (RC) - Title: Start with rotaviruses for the sake of consistency. Throughout the text, you always mention rotaviruses first.

Response: This was similar to R1 comment.  We agree with the opinion of the reviewers.  In addition, we have changed Table 1 to be rotavirus before norovirus.

RC - Lines 12-13: "This study examined the rates of viral infections throughout Ukraine" - you should mention which viral infections you intend to study

Response: We agree, changed.

RC - Line 23: you do not need to mention the national borders because it is confusing. For a non-Ukrainian, the relationship between the proximity to the borders and environmental issues is not apparent.

Response Changed

INTRODUCTION
RC - Please review the referencing style. MDPI's is specific, and you can find it in the author's guidelines.

Response – The senior author (Rodd) would like to apologize for the errors in the guidelines.  ZAR was submitting multiple manuscript to vary different journals (Journal of Neurochemistry, Neuroscience, and the journal Security). I made the errors in the MDPI format (slightly embarrassing because I just finished being the guest editor for the International Journal of Molecular Science (MDPI).  They have been corrected.

RC Other comments about Introduction.

Response – These comments have been addressed in the response to the other reviewers.

RC - MATERIAL AND METHODS
Lines 113-116: mention the number of the ethical approval letter for tracking purposes.

Response – There is something of a unique situation in Ukraine.  The 2nd Presidential period of Viktor Yanukovych (let’s not talk about the 2004 election in which Russian operatives poisoned Yanukovych’s opponent) occurred between 2010-2014 (ended with the Euromaidan Revolution.  The Yanukovych reign was seen as highly corrupt.  Yanukovych’s relatives and cronies gained control over the ‘for-profit’ scientific testing in the country (including the ethical approval of studies).  Between 2014-2017, most Ukrainian universities/centers for education were purged of the Yanukovych influence (please see the anti-corruption link).  During this time, all previously approved IRBs were voided at KPI and most other Ukrainian research facilities.  When the study was initiated (2016), KPI only had a code of honor approval process.  We are hoping that the reviewer understands our desire not to publicly detailed this period of Ukraine research.

https://kpi.ua/en/program-anticor

https://kpi.ua/en/code

RC – RESULTS Do not present the results in bulleted paragraphs. Use normal text.

Response – Totally the fault of Rodd.  Corrected.

RC - Figure 1's content will be better presented as two circular graphs with percentages. You can put them side by side as a figure with A and B.

Response – We examined the figures described by the reviewer, but we prefer the figures as is.

RC - I propose you present some graphs of co-infection (rotavirus-norovirus) if possible. That would be a piece of valuable information.

Response – Because of the war, we are not able to get the detailed data for the co-infection.  We agree with the reviewer that the co-infection data would be of high interest, but we cannot access the data right now.

RC - CONCLUSION
Line 358: instead of "poses" write "pose"

Response - corrected

RC - REFERENCES
There are some inconsistencies:
1. Use MDPI style
2. Why are some titles in capital letters?

Response – MDPI style is now used.

  1. I know this may seem odd, but the references were autogenerated by PubMed. For two articles the titles were all capitalized in PubMed. We were uncertain if that would be the proper manner to cite these articles.   We have corrected the CAPITALIZED REFERENCES.

Reviewer 6 Report

In the submitted manuscript, an epidemiological survey on rotavirus and norovirus in children hospitalized for diarrhea was provided. Overall, the poll results and discussion are convincing; nevertheless, substantial language editing would be necessary before publishing.

The results should be written in paragraphs rather than bullet points. This document is difficult to read due to missing words and typos.

Listed below are a few examples:

  • Line 69; "The first widely used rotavirus (Rotarix – RV1, GlaxoSmithKlein)"
  • Line 123; " The phosphate-salt buffer suspended stool samples to obtain 50% suspension of feces." 
  • Line 124; "The suspension was spilled into three sterile plastic test tubes (ependorfs)"

Author Response

Response to Reviewer 6

We would like to thank the reviewer for their encouragement and the useful comments that they provided us.  We have incorporated all of the reviewer’s comments into the manuscript.

Reviewer’s Comments - The results should be written in paragraphs rather than bullet points.

Response Corrected

RC - Method Section

Response – The Method section was directly translated from Ukrainian to English.  We have smoothed the English translation of the methods.

Round 2

Reviewer 1 Report

I would first like to recognize and commend the authors for their commitment to publishing this research during a very difficult time in Ukraine. The persistence and courage of the study team is admirable, especially given the uncertainty with regards to some colleagues.

Generally, the authors have addressed my comments. I have a few minor suggestions remaining.

  • Methods: Please add, perhaps in line 99, that clinical (symptom, treatment) and demographic information about participants was obtained from clinician records rather than from parental interview.
  • Methods Line 146 : Please add “(40/41)” after adenovirus, so that the reader can be sure which adenoviruses were tested for.
  • Overall, the results section is much clearer.
  • Since the data are not available to show the exact prevalence of astrovirus and adenovirus by age, I suggest adding a sentence in line 181 or in the Discussion stating that these data are not available.
  • Results line 183: I think you mean to say that the range was 0 – 36 months of age? (Not 0 – 16 months of age?)
  • Results line 184-185: please move the comparison to another study to the Discussion section
  • Results line 216: Do you mean to say that Lyiv was the only community where high prevalence was observed for both rotavirus and norovirus? As written, it sounds like you are speaking about co-infections, but I think that you do not presently have data on co-infections?
  • Discussion: suggest adding to the limitations the fact that some secondary analyses (e.g., detailed prevalence of astrovirus and adenovirus by age and other factors) were not able to be performed due to the situation in Ukraine making it impossible to access the data
  • Discussion: please add to the limitations that samples were not able to be collected year-round due to low parental acceptance of enrollment. This may have biased prevalence estimates upwards if enrollment coincided with peak seasons for viral gastroenteritis, or downward if enrollment coincided with low seasons for viral gastroenteritis. These results are thus only generalizable to the season during which the samples were collected.

Author Response

First, we thank the reviewer for their helpful suggestions.

We have performed all the suggestions from the reviewer.

Reviewer 2 Report

Most of comments made previouly were not addessed properly.

The overall quality of the manucript is low, although the subject of this study has some merits. 

Author Response

This is the second disappointing review from this individual.

In the first review, the comments were incorrect and defamatory. 

It is a sad commentary that the reviewer has responded with this superficial, spiteful review. 

We are glad that we had five professional reviewers for our manuscript.

Reviewer 3 Report

I would like to congratulate the authors on their work. The revised manuscript looks much better. The authors have addressed all my comments. 

Author Response

We thank the reviewer for their supportive comments.

Reviewer 6 Report

Thanks for the update.

Author Response

We thank the reviewer for their helpful suggestions and encouragement.